# Research on Void Dynamics during In Situ Consolidation of CF/High-Performance Thermoplastic Composite

**DOI:** 10.3390/polym14071401

**Published:** 2022-03-30

**Authors:** Qinghua Song, Weiping Liu, Jiping Chen, Dacheng Zhao, Cheng Yi, Ruili Liu, Yi Geng, Yang Yang, Yizhu Zheng, Yuhui Yuan

**Affiliations:** 1Composites Center, COMAC Shanghai Aircraft Manufacturing Co., Ltd., Shanghai 201324, China; songqinghua@comac.cc (Q.S.); chenjiping@comac.cc (J.C.); yicheng@comac.cc (C.Y.); liuruili@comac.cc (R.L.); gengyi@comac.cc (Y.G.); yangyang@comac.cc (Y.Y.); zhengyizhu@comac.cc (Y.Z.); yuanyuhui@comac.cc (Y.Y.); 2College of Material Science and Engineering, Donghua University, Shanghai 201620, China; 18817831097@163.com

**Keywords:** thermoplastic composites, automated fiber placement, in situ consolidation, void content, processing parameters

## Abstract

Automated fiber placement (AFP) in situ consolidation of continuous CF/high-performance thermoplastic composite is the key technology for efficient and low-cost manufacturing of large thermoplastic composites. However, the void in the in situ composite is difficult to eliminate because of the high pressure and the short consolidation time; the void content percentage consequently is the important defect that determines the performance of the thermoplastic composite parts. In this paper, based on the two-dimensional Newtonian fluid extrusion flow model, the void dynamics model and boundary conditions were established. The changes of the void content percentage were predicted by the cyclic iteration method. It was found that the void content percentage increased gradually along the direction of the layers’ thickness. With the increasing of the laying speed, the void content percentage increased gradually. With the increasing of the pressure of the roller, the void content percentage gradually decreased. When the AFP speed was 11 m/min and the pressure of the compaction roller reached 2000 N, the void content percentage of the layers fell below 2%. It was verified by the AFP test that the measured results of the layers’ thickness were in good agreement with the predicted results of the model, and the test results of the void content percentage were basically equivalent to the predicted results at different AFP speeds, which indicates that the kinetic model established in this paper is representative to predict the void content percentage. According to the metallographic observation, it was also found that the repeated pressure of the roller was helpful to reduce the void content percentage.

## 1. Introduction

Automated fiber placement (AFP) in situ consolidation of continuous CF/high-performance thermoplastic composite (Figure 1) [1,2,3,4,5], which is based on continuous fiber-reinforced thermoplastic prepreg, positioning, laying-up, and consolidation by automatic placement equipment according to the requirements of the mathematical model, components are manufactured in the thickness direction by layer accumulation, and the manufacturing is accomplished when the thickness reaches the designed value; thus, the traditional manufacturing process of first laying-up and then curing in the autoclave is no longer needed [6,7,8,9,10,11,12]. This technology will bring about an unprecedented change in composite components’ manufacturing, especially in the aerospace applications of composite materials.

The thermoplastic composite has outstanding characteristics, which are different from the traditional thermosetting composite, such as high toughness, high strength, fast forming, and recyclability [13,14,15,16,17,18]. It is a new type of composite material with high performance, low cost, and environmental protection and becomes an ideal material for civil aircraft structure components manufacturing, which can reduce weight effectively and improve processing efficiency [19,20]. Airbus, Boeing, etc., have developed special AFP equipment, which was applied to manufacture the aircraft secondary bearing structure.

Although the application of thermoplastic composite materials in civil aircraft has gradually aroused an upsurge in research upsurge abroad and in the future, the large-scale application of thermoplastic composite materials in civil aircraft is the inevitable result of the development of advanced materials and the progress of manufacturing technology. Nevertheless, the continuous fiber-reinforced thermoplastic composite in situ consolidation technology has not been applied in the manufacturing of civil aircraft structural components effectively at present [21,22]. The reason is the significant difference between the AFP in situ consolidation process of thermoplastic composite and the AFP of thermosetting composite. The AFP of thermosetting composite is a process of shaping, only completing the pre-forming of structural components, and the forming is completed mainly in the autoclave. Despite the fact that the process of AFP in situ consolidation of thermoplastic composite is very complex, the high melting point and melting viscosity of the high-performance thermoplastic resin matrix requires high temperature and pressure; thus, the requirements for manufacturing equipment are more demanding.

In addition, the residual air bubbles or dissolved air, as well as water or other volatiles will lead to an increase of the void content of the structural components manufactured by in situ consolidation. Void content is one of the most important defects that determines the structural component properties of thermoplastic composites. Consequently, it is necessary to reduce the void content of the layers as much as possible, to improve the mechanical properties [23,24,25,26].

The void content of thermoplastic composites has been studied by many scholars. Qinghao He [27] quantified the adverse effects of voids on 3D-printed continuous fiber-reinforced polymer composites. Vipin Kumar [28] produced a large-scale multimaterial by additive manufacturing (AM) undergoing compression molding (CM) to produce high-performance thermoplastic composites reinforced with short carbon fibers. Zhu Liu [29] established a microscale unit cell with a random distribution of fibers, interfaces, and voids based on the random sequential adsorption algorithm to investigate the quantitative effects of void content on the strength and modulus under the loading of transverse tension.

In the process of in situ consolidation, the prepreg and the substrate are melted in the bonding area under the heating of the heat source, and pressure is applied through the compaction roller, then the layers are bonded together in the bonding area and finally cooled and consolidated simultaneously, to realize the in situ consolidation. As a result, the void content due to the reasons mentioned above can be reduced by increasing the pressure under the compaction roller. In the forming process of thermosetting composite, due to the low viscosity of the resin matrix, Darcy’s law can be used to describe the process of resin impregnation. In contrast, the melting viscosity of the thermoplastic resin matrix is high, and the resin and the fiber are not relatively independent during the forming process; they generally move with the external pressure [30]. Hence, Balasubramanyam [31] and Barnes [32] thought that the fiber, resin, and voids in fiber-reinforced thermoplastic composite can be approximated as a uniform continuum, and this is a more valuable way to describe the flow of the resin as the flow of the continuum in the process of forming. Tierney [33] also pointed out that, compared with Darcy’s law model, they believed that the squeeze flow model was more suitable for analyzing the change of void content in the processing of thermoplastic composites. Muhammad [34] established a two-dimensional Newtonian fluid squeeze flow model for the processing of thermoplastic composites.

In this paper, the objective of this work was establishing a two-dimensional Newtonian fluid squeeze model to describe the void dynamics during the in situ consolidation process. The variety of the void content with the processing parameters during the in situ consolidation process was predicted through the cyclic iterative method.

## 2. Void Dynamics Model

### 2.1. Void Formation Mechanisms

In the in situ consolidation process, voids are usually inter-laminar and intra-laminar. The intra-laminar voids mainly contribute to the following aspects: (1) softening of the resin matrix, (2) primary voids in the prepreg, and (3) dissolved air. The inter-laminar voids are mainly caused by the fact that the intimate contact between layers has not reached to one (Figure 2) [35]; hence, there exists residual air between the layers.

During the in situ consolidation process, the prepreg and the substrate are melted by the heat source first and bonded together by the compaction roller, and then, the bonded layers are cooled and crystallized. Before the pressure is applied, the resin matrix of the prepreg experiences softening, thermal expansion, and melting in a short heating time; at the same time, the dissolved volatiles and the air remaining in the prepreg are released, and the voids in the layer begin to appear simultaneously. Subsequently, the pressure of the compaction roller plays a significant role in both preventing the voids’ growth and eliminating the inter-laminar and intra-laminar voids (Figure 3).

In Figure 3, the initial thickness of the layers is $h_{i}$, the width is $w_{i}$, x is the width direction, y is the direction of movement, z is the thickness direction, and the placement speed is v. As the compaction roller passes through the bonding area, the thickness of the layer is reduced to $h_{f}$ and the width changes to $w_{f}$.The consolidation pressure compresses the voids and changes the thickness and width of the layers. Since the layer dimension in the y direction is much larger than the x and z directions, the y direction flow may be neglected. Due to the high matrix resin viscosity, the flow may be treated as creeping motion. In other words, the inertial effects can be neglected. Since the y direction flow can be neglected, the in situ consolidation process can be treated as a sequence of two-dimensional squeeze flow problems in the x−z plane. The squeeze speed in the z direction is governed by the placement speed v.

From Figure 3, the contact length $L_{c}$ between the compaction roller and prepreg can be described as:(1)$$L_{c}={\left[R_{r}^{2}-{\left(R_{r}-h_{i}+h_{f}\right)}^{2}\right]}^{1/2}$$

Similarly, the height h of the layer may be expressed as:(2)$$h=R_{r}+h_{f}-{\left[R_{r}^{2}-{\left(L_{c}-y_{c}\right)}^{2}\right]}^{1/2}$$
where $y_{c}$ represents the distance between the contact point of the prepreg and the front of the compaction roller (the position where the thickness of the layer is h).

Differentiating both sides with respect to time t and equating $dy_{c}/dt$ to the placement velocity v, we can obtain:(3)$$\overset{\cdot }{h}=\frac{dh}{dt}=-v\frac{{\left[R_{r}^{2}-{\left(R_{r}-h+h_{f}\right)}^{2}\right]}^{1/2}}{\left(R_{r}-h+h_{f}\right)}$$

Equation (3) is an expression of the closing speed $\overset{\cdot }{h}$ at the layer interface.

Besides, the width w of the prepreg during in situ consolidation can be obtained by the average speed of the free boundary condition of the prepreg, which can be expressed as:(4)$$\frac{dw}{dt}=\frac{1}{h}{\left[\int_{0}^{h}v_{x}dz\right]}_{x=w}$$
where $v_{x}$ is the speed of the squeeze flow of the resin matrix under the pressure of the compaction roller.

By integrating the contact length between the compaction roller and the prepreg and the real-time width of the prepreg, the pressure on the layers during the in situ consolidation process can be determined as:(5)$$F_{c}=2\int_{0}^{L_{c}}\int_{0}^{w}P(x,y)dxdy$$
where P(x,y) is the pressure distribution under the compaction roller.

Therefore, in this work, the in situ consolidation process was simplified as a two-dimensional Newtonian fluid squeeze flow process under the pressure provided by the compaction roller. In this process, the changes of the layer thickness and width were considered, and the change of the layer length was neglected. Simultaneously, the changes of the height and width were related to the squeeze flow velocity of the resin and the pressure distribution under the compaction roller. Thus, the macro void dynamics model, containing the squeeze flow model of the resin and the pressure distribution model, is proposed. To solve the model, the velocity boundary condition and the pressure boundary condition were also considered. Finally, the void dynamics model was solved by the cyclic iteration method to predict the variety of layer void content during the in situ consolidation process. The flowchart of the proposed models is shown in Figure 4.

### 2.2. Void Dynamics Model

The momentum equations that govern the flow of the continuum under the compaction roller, neglecting the inertia and body terms, can be written as:(6)$$\frac{\partial \rho }{\partial t}+\frac{\partial }{\partial x}\left(\rho v_{x}\right)+\frac{\partial }{\partial z}(\rho v_{z})=0$$
(7)$$\frac{\partial P}{\partial x}=\frac{\partial }{\partial z}\left(\mu \frac{\partial v_{x}}{\partial z}\right)$$
where the density of the continuum is given by ρ, the viscosity of the resin matrix by μ, the compaction roller pressure by P, and the placement velocity by *v*.

By integrating Equation (7) with respect to z:(8)$$\mu [\frac{\partial v_{x}}{\partial z}]=z\frac{dP}{dx}+C_{1}(x)$$

Integrating Equation (8) once more with respect to z, the flow velocity model of resin squeeze during in situ consolidation can be obtained:(9)$$v_{x}(z)=v_{x}(0)+\frac{dP}{dx}\int_{0}^{z}\frac{\xi }{\mu }d\xi +C_{1}(x)\int_{0}^{z}\frac{1}{\mu }d\xi$$

Integrating Equation (6) in the thickness direction from 0 to *h*, the governing equation describing the instantaneous thickness of the layer can be obtained:(10)$$h\frac{\partial \rho }{\partial t}+\int_{0}^{h}\left[\frac{\partial }{\partial x}\left(\rho v_{x}\right)\right]dz+\rho \left(v_{z}\right)|{}_{z=0}^{z=h}=0$$

The velocity $v_{z}$ at z=0 is 0 and at z=h is close to the closing speed between the two bounding surfaces, so assuming $v_{z}=\overset{\cdot }{h}$. Thus, substituting variables and changing the order of integration and differentiation in Equation (10), the following expression can be obtained:(11)$$h\frac{\partial \rho }{\partial t}+\frac{\partial }{\partial x}\left(\rho \int_{0}^{h}v_{x}dz\right)+\rho \overset{\cdot }{h}=0$$

Substituting the velocity variable in Equation (9) into Equation (11) to eliminate the velocity variable:(12)$$h\frac{\partial \rho }{\partial t}+\frac{\partial }{\partial x}\left(\rho \int_{0}^{h}\left[v_{x}(0)+\frac{dP}{dx}\int_{0}^{z}\frac{\xi }{\mu }d\xi +C_{1}(x)\int_{0}^{z}\frac{1}{\mu }d\xi \right]dz\right)+\rho \overset{\cdot }{h}=0$$
where h is the instantaneous thickness of the layer; $C_{1}\left(x\right)$ and $v_{x}\left(0\right)$ can be obtained from two velocity boundary conditions. Because Equation (12) is a second-order integral equation, so two pressure boundary conditions are also needed.

Defining a non-dimensional density $\rho^{*}$ of layer:(13)$$\rho^{*}=\frac{\rho }{\rho_{f}}$$
where $\rho^{*}$ is the initial density of prepreg. Therefore, Equation (12) can be modified as:(14)$$h\frac{\partial \rho^{*}}{\partial t}+\frac{\partial }{\partial x}\left(\rho^{*}\int_{0}^{h}\left[v_{x}(0)+\frac{dP}{dx}\int_{0}^{z}\frac{\xi }{\mu }d\xi +C_{1}(x)\int_{0}^{z}\frac{1}{\mu }d\xi \right]dz\right)+\rho^{*}\overset{\cdot }{h}=0$$

Equation (14) is considered as the governing equation of the pressure distribution model under the compaction roller during in situ consolidation.

### 2.3. Boundary Conditions

Two velocity boundary conditions and two pressure boundary conditions are needed for the solution of the void dynamics model.

#### 2.3.1. Velocity Boundary Conditions

Barone J.R. [36] established a velocity boundary condition between the fluid and solid in the compression molding process, since the contact between the prepreg and the compaction roller during the in situ consolidation process is similar to compression molding; therefore, based on Barone’s theory, the velocity boundary condition in this work was described by a friction factor boundary condition and expressed as:(15)$$\frac{\partial v_{x}}{\partial z}=\frac{K}{\mu }{\left(v_{x}\right)}_{z=h}$$

The ratio $\frac{K}{\mu }$ determines the type of boundary condition. If the friction at the layer interface is very high, i.e., $\frac{K}{\mu }\to \infty$, this is equivalent to a no-slip boundary condition, then ${\left(v_{x}\right)}_{z=h}$ is expressed as:(16)$${\left(v_{x}\right)}_{z=h}=0$$

On the other hand, if the friction is expected to be very low, i.e., $\frac{K}{\mu }\to 0$, this is equivalent to a perfect boundary condition:(17)$$\frac{\partial v_{x}}{\partial z}=0$$

#### 2.3.2. Pressure Boundary Condition

If the layers are assumed to be unrestricted along the width direction, then the appropriate boundary condition would be to impose atmospheric pressure at the free surfaces, which could be expressed as:(18)$$P_{\mathrm{edges}}=P_{\mathrm{atm}}$$

Since the width of the compaction roller is not less than the layer, so it was assumed that the pressure of compaction roller along the layers’ width direction is uniformly applied.

## 3. Materials and Methods

### 3.1. Material

The material selected in this paper was CF/PPS prepreg with a width of 6.35 mm. Relevant parameters required for the model simulation are given in Table 1. Figure 5 is the metallographic micrograph (Metalloscope, OLYMPUS, Tokyo, Japan) of the void content of the initial prepreg. It could be calculated that the initial void content of the prepreg was about 0.8% by the metallographic observation.

### 3.2. Model Prediction Methods

Figure 6 is the computational flowchart of void dynamics model prediction. Before prediction, the ratio $h_{f}/h_{i}$ of the layer compressed by the roller can be assigned an initial value, and a roller pressure $F_{cr}$ was applied. Then, the geometric parameters of the layer under the pressure of the compaction roller can be obtained according to Equations (1) and (2); the change in thickness of the layers can be derived from Equation (3); the squeeze flow speed of the resin matrix along the width direction can be predicted using Equation (9); accordingly, combined with Equation (4), the variety of prepreg width in the process of in situ consolidation was obtained; the variety of the layer in the y direction was neglected, that is assuming the prepreg dimension is constant along the length direction. Therefore, the changes of the layer volume in the process of in situ consolidation can be predicted by combining the change of the layer width and thickness. In addition, the resin matrix was assumed to be incompressible, so according to the law of the conservation of mass, the density of the layer can be obtained. Since the density and void content of the original prepreg are known, the change in the void content of the layer during the in situ consolidation process can be solved.

The pressure distribution under the compaction roller can be obtained by Equation (14), and then, combined with Equation (5), the theoretical roller pressure $F_{c}$ can be calculated. Comparing the theoretical roller pressure $F_{c}$ with the actual roller pressure $F_{cr}$, if they are not equal, $h_{f}/h_{i}$ will be reassigned and recalculated until $F_{c}=F_{cr}$. When the roller leaves the bonding area, the layers are exposed to the air, and the real-time variety of layers’ void content and density can still be solved by the void dynamics model. The method mentioned above was used to predict the void content of each layer.

### 3.3. Experimental Section

#### 3.3.1. Thickness Measurement

Three specimens with a size of 400 × 400 mm^2^ were manufactured at different placement parameters to verify the void dynamics model, as shown in Table 2. The prepreg was unidirectional carbon-fiber-reinforced polyphenylene sulfide composite (CF/PPS). Six layers of prepreg were laid on the flat tooling under the 4 tows of the AFP equipment shown in Figure 7 (M·Torres, Navarra, Spain). The main purpose of this test was to verify the effect of different placement speeds on the void content of the layer.

During the manufacturing process of the specimens, the thickness vernier caliper (Mitutoyo, Kawasaki City, Japan) was used to measure the thickness of each layer of prepreg after placement, as shown in Figure 8.

#### 3.3.2. Metallographic Observation

Metallographic observation samples were prepared by cutting, inlaying, grinding, polishing, and other processes according to the GB 3365-2008 standard (Figure 9).

## 4. Results and Discussion

### 4.1. Model Prediction

Figure 10 shows the through-thickness void content distribution with the change of the placement speed through model prediction. The compaction roller pressure was set to 1000 N, the laser power to a maximum value of 6 kW, and the placement speed to 11 m/min, 13 m/min, and 15 m/min, respectively.

It can be seen from Figure 10 that placement speed had a significant effect on the void content. The results of the temperature field [37] and inter-laminar bonding strength analysis [35] indicated that the consolidation of the layers resulted in a combination of temperature and pressure. On the one hand, at higher velocities, the heat input was significantly reduced, and the insufficient heat led to a high resin viscosity, which restricted the elimination of the voids; on the other hand, a higher placement speed reduced the pressure application time, leading to a higher void content.

Figure 10 also indicates that the void content of the bottom layer and the upper layers along the thickness direction was slightly higher. On account of the direct contact between the first layer and the mold, resulting in the lower heating temperature that caused the high resin viscosity, the void content was consequently high. When laying the second layer, the temperature increased and the resin viscosity decreased due to the heat insulation of the first layer, so the void content decreased. However, due to the heat accumulation effect of the layer, the temperature of the layers gradually increased with the increase of the layers’ thickness, which was responsible for the extremely high temperature of the layers even if the roller left the bonding area. Furthermore, if the temperature of the layers was still above the melting point, the molten resin would dissolve the air, and thus, the void content would increase.

Figure 11 shows that the void content varied with the laser power and compaction roller pressure. The placement speed was set to 11 m/min.

It can be seen from Figure 11 that the void content was relatively high when the pressure under the compaction roller was small. The reason lied in that during the process of in situ consolidation, due to the resin being in a molten state in the bonding area when the pressure roller had not passed through, the voids inside the layer were not constrained by external pressure and began to expand, resulting in a higher void content; when the roller passed through the bonding area, the pressure under the compaction roller was not enough to discharge most of the voids; on the other hand, due to the high pressure inside the voids, the binding force of the voids decreased when the roller left the bonding area, so the voids began to expand, leading to a higher void content. The result in the figure also indicates that a lower the layer void content could be obtained by increasing the pressure under the compaction roller; however, the layer void content could be reduced to less than 2% only when the pressure under the compaction roller reached 2000 N or higher. In addition, increasing the heating power reduced the effect of pressure on the void content. This was due to the viscosity of the resin increasing due to sufficient heat input at a high laser power, which was beneficial to eliminate the voids; additionally, it is manifest from the figure that the void content of the layers at a maximum laser power of 6 kW and a small roller pressure was still greater than that at a heating power of 4 kW with larger roller pressure, which indicates that the void content can be attributed to the coupling effect of the laser power and the pressure under the pressure roller.

### 4.2. Experimental Verification

Figure 12 shows the results of the thickness measurement, and the straight line in the figure shows the simulation results. It can be seen that the thickness of each layer was stable, and the simulation results of the model were in good agreement with the measured values.

Figure 13 shows images of the void content of the specimens manufactured by the process parameters shown in Table 2. It is obvious that void content of the layers gradually increased with the increasing placement speed. Nonetheless, none of the three figures show the trend of higher void content in the bottom layer, as shown in Figure 10. The reason for the preliminary analysis was that the first layer in Figure 10 was only rolled once by the compaction roller, so the void content of the first layer was high. On the contrary, with the increasing of the number of layers, the number of times that the bottom layer prepreg was rolled increased. After the repeated pressure of the roller, the void content of the bottom layer accordingly decreased, whereas the number of rolls on the upper layer reduced less gradually; therefore, the void content increased gradually with the increase of the number of layers. The results of the metallographic test thereby were similar to the simulation results shown in Figure 6, except for the bottom prepreg.

Figure 14 shows the comparison between the void content test results of the specimens manufactured by different placement speeds and the simulation results of the model. It is obvious that the void content showed an increasing trend when the placement speed increased. It can also be seen that the test results were equivalent to the simulation results, which proves that the dynamics model established in this work had a certain representativeness for predicting the void content of the layer, but the prediction results were always slightly lower than the measurement results, which was related to the simplification of the model, such as the change of the layer along the length direction being neglected; however, the fact is that during the in situ consolidation process, the layer would become longer along the length direction, which would result in a higher predicted density; thereby, the void content obtained by the model was slightly lower.

## 5. Conclusions

A void dynamics model and boundary conditions during the process of in situ consolidation were established based on the two-dimensional Newtonian fluid squeeze flow model. The cyclic iteration method was used to predict the variety of the void content during the process of in situ consolidation. The experimental measurement results of the layers’ thickness were in good agreement with the prediction results, and the void content test results of the specimens manufactured at different placement speeds were equivalent to the prediction results, which means the void dynamics model established in this work has certain representativeness for predicting the void content; the metallographic observation results also indicated that the repeated pressure of the compaction roller was beneficial to reduce the void content.

## Figures and Tables

**Figure 1 polymers-14-01401-f001:**
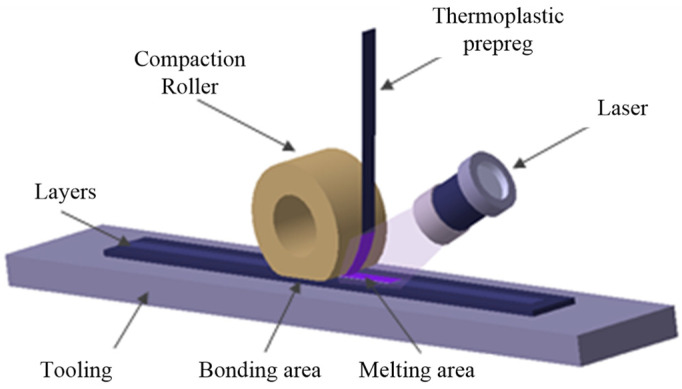
Schematic diagram of AFP in situ consolidation.

**Figure 2 polymers-14-01401-f002:**
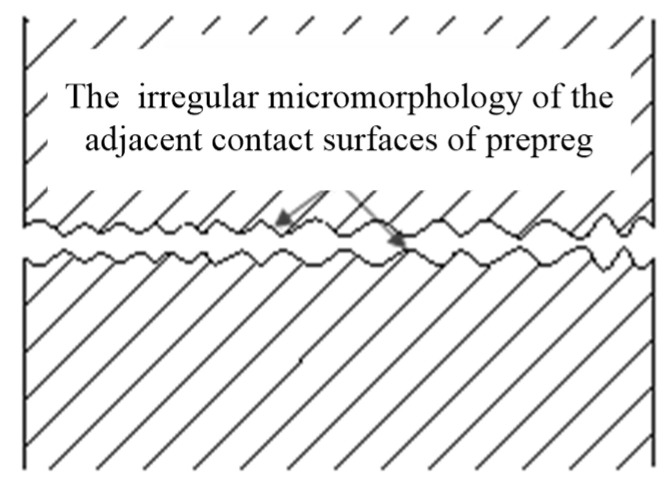
Schematic diagram of the contact between two layers.

**Figure 3 polymers-14-01401-f003:**
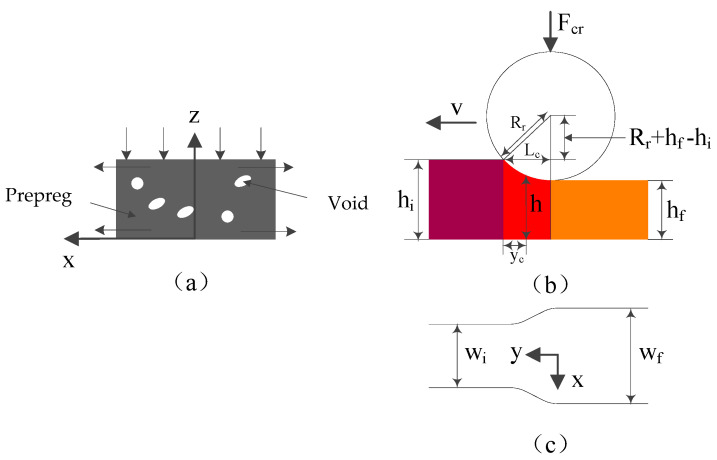
Geometric sketch of the in situ consolidation process: (**a**) The thermoplastic prepreg layer; (**b**) The pressure is applied by compaction roller; (**c**) The prepreg is widened under the pressure.

**Figure 4 polymers-14-01401-f004:**
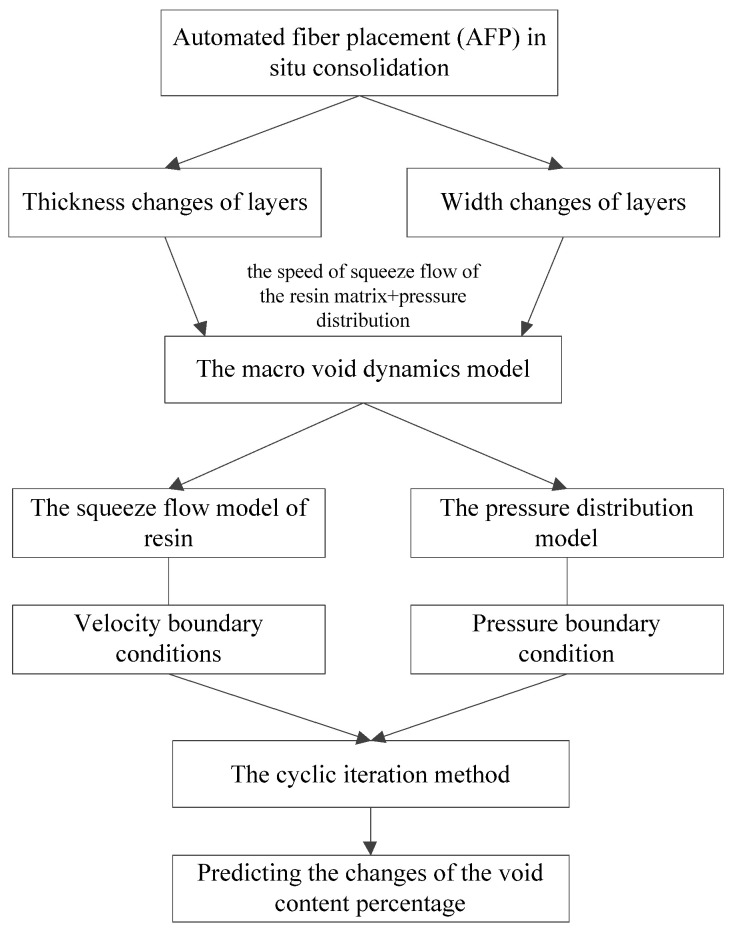
The flowchart of the proposed models.

**Figure 5 polymers-14-01401-f005:**
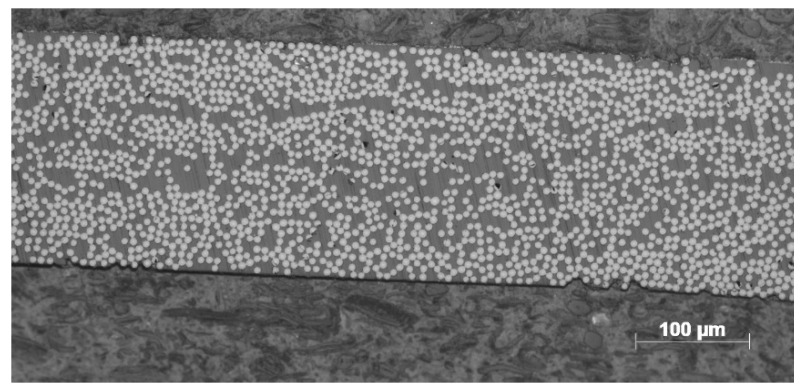
Metallographic micrograph of initial prepreg.

**Figure 6 polymers-14-01401-f006:**
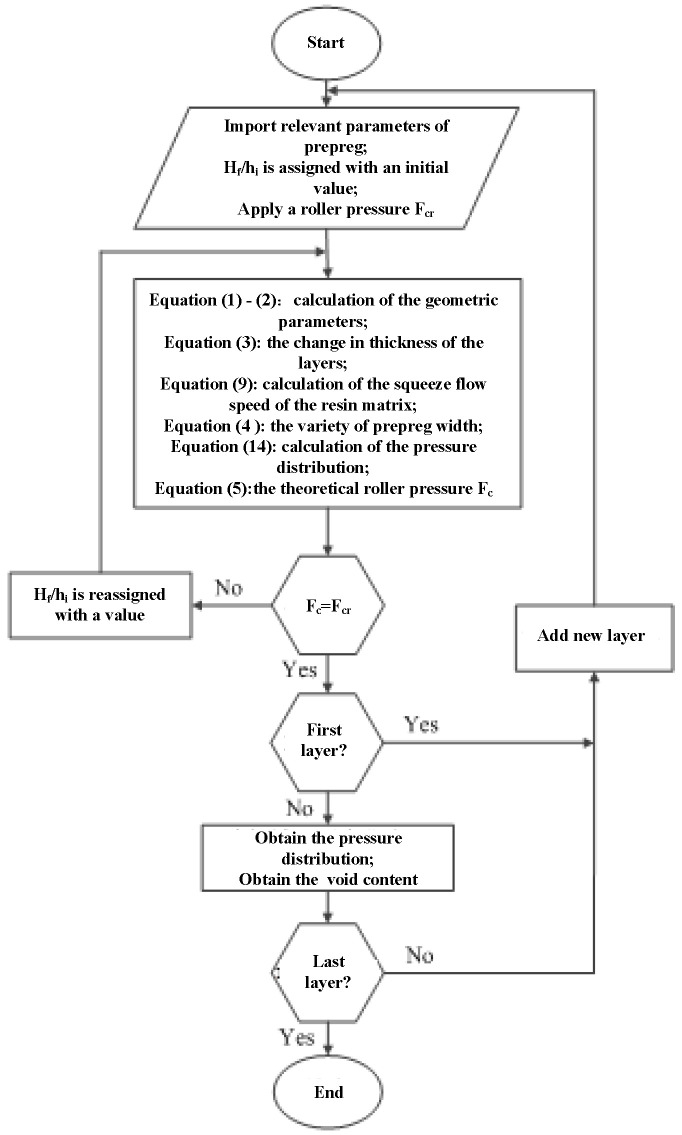
Calculation flowchart of the void dynamics model.

**Figure 7 polymers-14-01401-f007:**
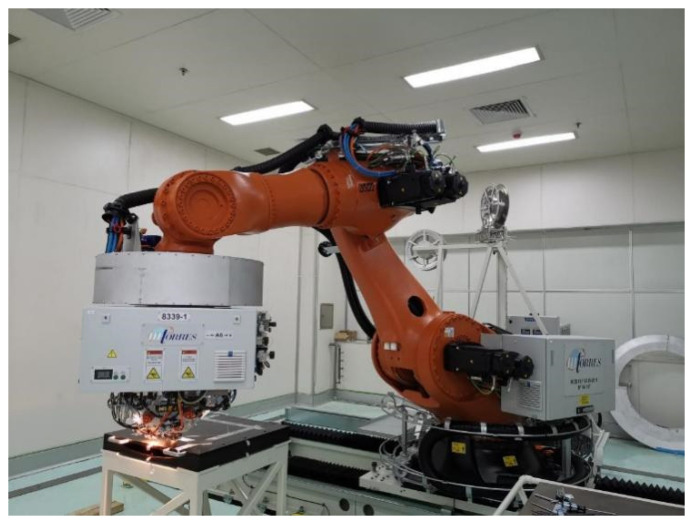
The equipment 4 tows of the AFP.

**Figure 8 polymers-14-01401-f008:**
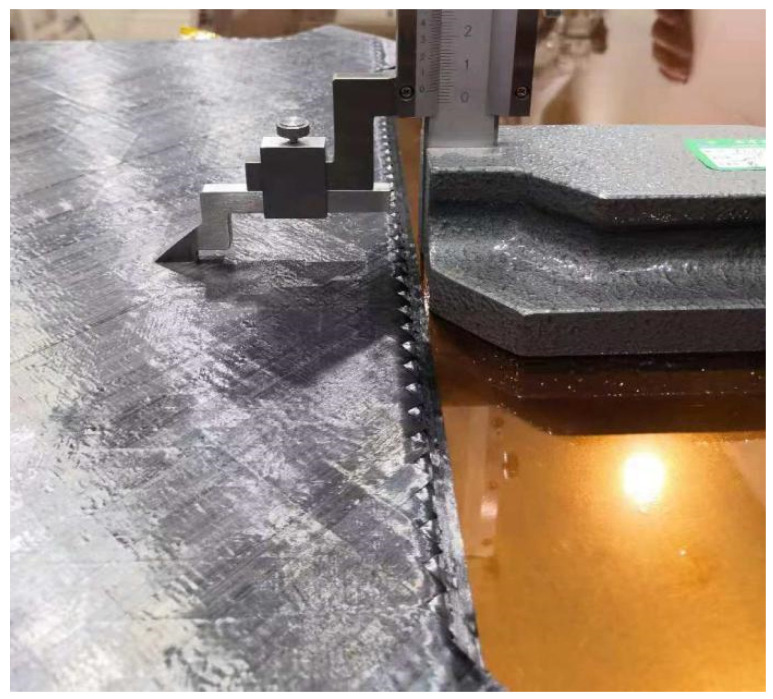
The thickness vernier caliper.

**Figure 9 polymers-14-01401-f009:**
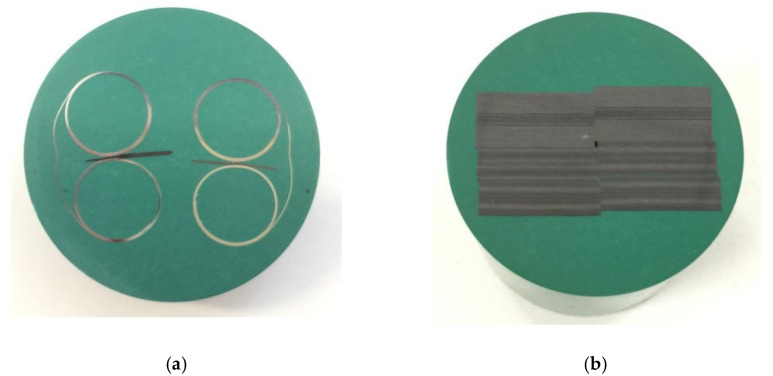
The metallographic observation samples: (**a**) one layer; (**b**) six layers.

**Figure 10 polymers-14-01401-f010:**
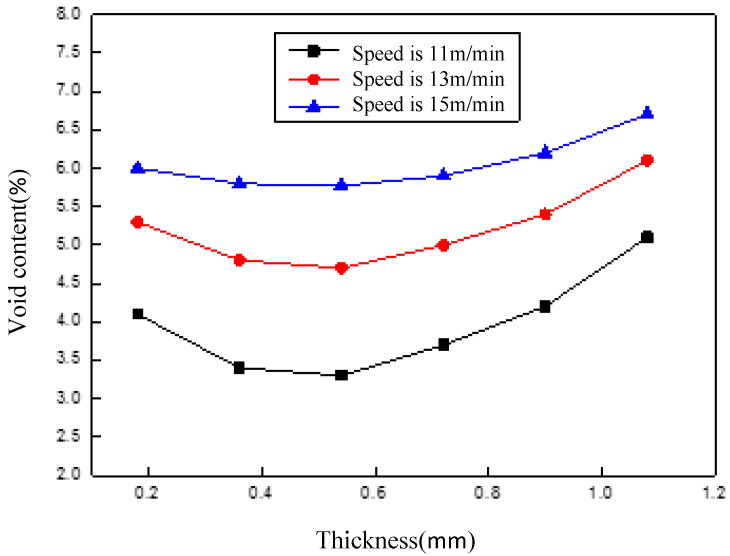
Through-thickness void content distribution.

**Figure 11 polymers-14-01401-f011:**
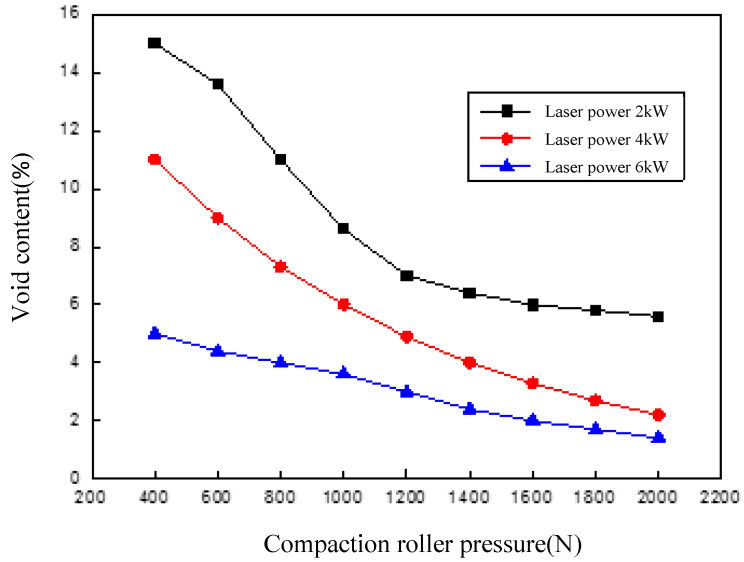
Void content varies with laser power and compaction roller pressure.

**Figure 12 polymers-14-01401-f012:**
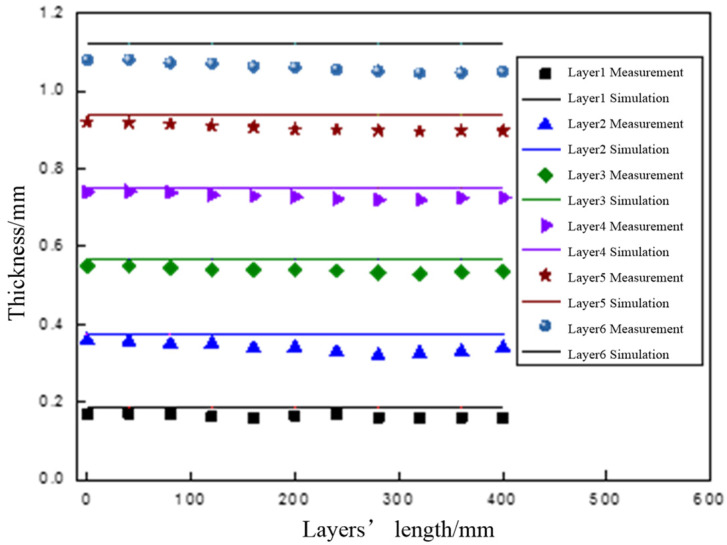
The comparison of the measured values and predicted values (speed is 11 m/min).

**Figure 13 polymers-14-01401-f013:**
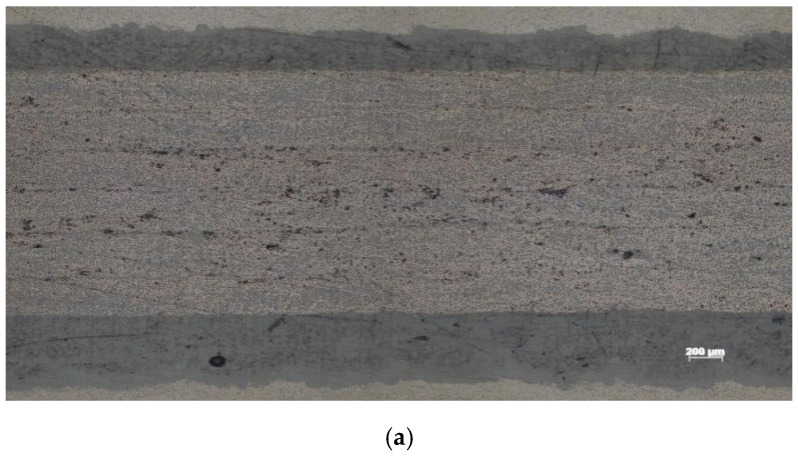
The images of the void content of the specimens manufactured at different speeds (laser power is 6 kW, pressure is 1000 N): (**a**) speed is 11 m/min; void content is 3.1%; (**b**) speed is 13 m/min; void content is 4.2%; (**c**) speed is 15 m/min; void content is 5.5%.

**Figure 14 polymers-14-01401-f014:**
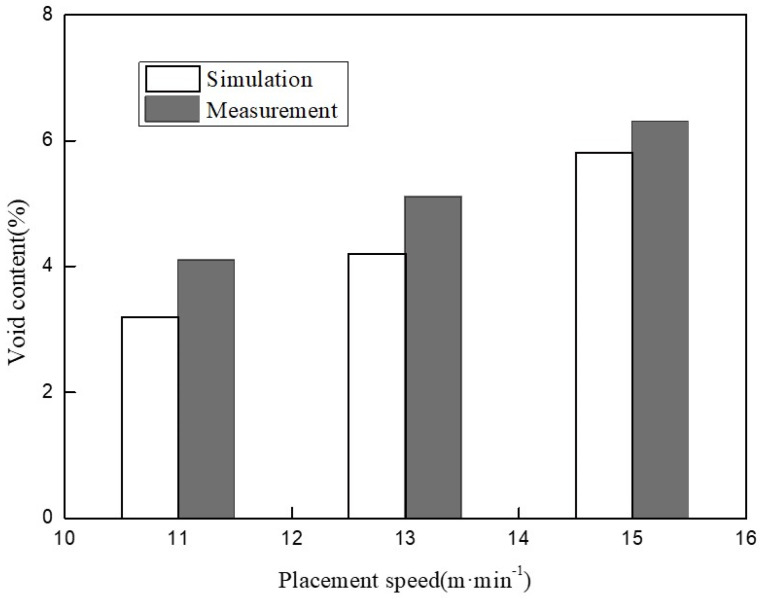
The void content varies with placement speed (laser power is 6 kW, pressure is 1000 N).

**Table 1 polymers-14-01401-t001:** Relevant parameters required for the model prediction.

Parameters	Symbol	Value
Number of layers	N	6
Compaction pressure, N	$F_{\mathrm{cr}}$	500–1000
Density of prepreg, g·cm^−3^	ρ	1.62

**Table 2 polymers-14-01401-t002:** Manufacturing parameters of the specimens.

No.	Placement Speed (m·min^−1^)	Laser Power (kW)	Pressure (N)
1	11	6	1000
2	13	6	1000
3	15	6	1000

## Data Availability

The data presented in this study are available upon request from the corresponding author.

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
