# Peer review of "Research on Void Dynamics during In Situ Consolidation of CF/High-Performance Thermoplastic Composite"

_polymers, 2022, doi:10.3390/polym14071401_

Round 1

Reviewer 1 Report

The paper entitled "Research on Void Dynamics during In-situ Consolidation of CF/High Performance Thermoplastic Composites" by Qinghua Song and co,  a void dynamic model was established for AFP in-situ consolidation 
which is based on the two-dimensional Newtonian fluid squeeze flow model to study the influence of the laying process parameters on the void content of the layers.

The paper needs some minor revisions, as follows:

1) Please chek again the format of the the enire paper (ex.: 2. Void formation mechanisms of layers is italic and 1. Introduction is bold)

2) In the Conclusion part, it is neccesary to highlight the novelty and the originality of the paper apart of reiterating the obtained results.

Author Response

Dear editor and reviewers:

Thank you for your comments. These comments help us to improve the quality of manuscripts. We have modified the manuscript according to each suggestion, and highlighted the modified part in the manuscript. Let's list the responses to each comment.

Thank you for your help on this manuscript.

Best wishes

Qinghua Song

Reply to Reviewers:

1 Please check again the format of the entire paper (ex.: 2. Void formation mechanisms of layers is italic and 1. Introduction is bold).

Reply: Thank you for your comments. The title of the chapters in the paper had been changed to standardized form and bold.

2 In the Conclusion part, it is neccesary to highlight the novelty and the originality of the paper apart of reiterating the obtained results.

Reply: Thank you for your comments. The novelty of this paper is that based on the two-dimensional Newtonian squeezing flow model, the void dynamics model and boundary conditions during the in-situ consolidation process were established, and the cyclic iteration method was used to calculate the void content; meanwhile, the model was verified through experiment that the dynamic model established in this paper was representative in predicting the void content during in-situ consolidation process. The innovation of this paper has been described in the Section 5 Summary and Conclusion, and the repeated experimental results have been deleted.

Reviewer 2 Report

The authors developed a hybrid model composed of the heat transfer model and the void dynamics model to calculate  the changes of the void content during  In-situ consolidation of CF/High performance thermoplastic composite. It is found that the void content percentage increases gradually along the direction of layers’ thickness. With the increase of laying speed, the void content percentage increases gradually. With the increase of the pressure of the roller, the void content percentage gradually decreases. When the AFP speed is 11m/min and the pressure of the compaction roller reaches 2000N, the void content percentage of the layers will fall below 2%. It is verified by AFP test that the measured results of layers’ thickness are in good agreement with the calculated results of the model, and the test results of void content percentage are basically equivalent to the calculated results at different AFP speeds, which indicates that the kinetic model established in this paper is representative in predicting the void content percentage. According to the metallographic observation, it is also found that the repeated pressure of the roller is helpful to reduce the void content percentage.

The paper introduces a valuable content and could be published after major revision.

I will be honored to review a revised version of the present paper before publication in polymers.

  1. The paper should be restructured and written in a clear manner using a high scientific style.
  2. English should be improved and checked by a native English speaker.
  3. Literature review should be improved.
  4. Missing research gap and the objectives of the paper should be clearly presented.
  5. Structure of the proposed models should be presented (using flow chart) and discussed.
  6. Section result and discussion need to be deeply improved.
  7. During the in-situ consolidation process, the prepreg and the substrate are melted by the heat source first, and then consolidated by the compaction roller. Why?
  8. The literature should be strengthen using recent publication in Polymers.
  9. Support the introduction using; fabrication techniques of polymeric nanocomposites: a comprehensive review, modeling of drilling process of gfrp composite using a hybrid random vector functional link network/parasitism-predation algorithm, effect of surface preparation on the strength of vibration welded butt joint made from pbt composite.
  10. A void dynamic model and boundary conditions during the process of in-situ consolidation were established based on the two-dimensional Newtonian fluid squeeze flow model. The cyclic iteration method was used to calculate the variety of the void content during the process of in-situ consolidation by combining the heat transfer model with the void dynamic model. This claim should be discussed.

Author Response

Dear editor and reviewers:

Thank you for your comments. These comments help us to improve the quality of manuscripts. We have modified the manuscript according to each suggestion, and highlighted the modified part in the manuscript. Let's list the responses to each comment.

Thank you for your help on this manuscript.

Best wishes

Qinghua Song

Reply to Reviewers:

  1. The paper should be restructured and written in a clear manner using a high scientific style.

Reply: Thank you for your comments. We have modified the manuscript structure as recommended.

2 English should be improved and checked by a native English speaker.

Reply: Thank you for your comments. We have revised the whole manuscript carefully and tried to avoid any grammar or syntax error. In addition, we have asked several colleagues who are skilled authors of English language papers to check the English. We believe that the language is now acceptable for the review process.

3 Literature review should be improved.

Reply: Thank you for your comments. Literature review have been improved. The lasted research was added and described to support the Section 1 Introduction.

The void content of thermoplastic composites has been studied by many scholars. Qinghao He[27]quantified the adverse effects of voids on 3D printed continuous fibre-reinforced polymer composites. Vipin Kumar[28] produced the large-scale multimaterial preforms by additive manufacturing (AM) underwent compression molding (CM) to produce high-performance thermoplastic composites reinforced with short carbon fibers. Zhu Liu[29] established a microscale unit cell with random distribution of fibers, interfaces and voids based on the random sequential adsorption algorithm to investigate The quantitative effects of voids content on strength and modulus under the loading of transverse tension.

4 Missing research gap and the objectives of the paper should be clearly presented.

Reply: Thank you for your comments. The objective of this work was establishing a two-dimension Newtonian fluid squeeze model to describe the void dynamic during the in-situ consolidation process. The variety of void content with processing parameters during the in-situ consolidation process was predicted through cyclic iterative method. The research gap and the objectives of the work have been presented in the Section 1 Introduction. 

5 Structure of the proposed models should be presented (using flow chart) and discussed.

Reply: Thank you for your comments. The structure of the proposed models and has been presented using a flow chart and detailed discussed in Section 2.1 Void formation mechanisms. 

In this work, the in-situ consolidation process was simplified as a two-dimensional Newtonian fluid squeeze flow process under the pressure provided by the compaction roller. In this process, the change of the layer thickness and width were considered, and the change of the layer length was neglected. Simultaneously, the changes of height and width were related to the squeeze flow velocity of resin and the pressure distribution under the compaction roller. Thus, the macro void dynamics model, containing the squeeze flow model of resin and pressure distribution model, was proposed. To solve the model, the velocity boundary condition and the pressure boundary condition were also considered. Finally, the void dynamic model was solved by cyclic iteration method to predict the variety of layer void content during the in-situ consolidation process. The flow chart of the proposed models was shown in Figure 2.

Fig.2 The flow chart of the proposed models

6 Section result and discussion need to be deeply improved.

Reply: Thank you for your comments. The structure of the manuscript has been modified and the result and discussion has been improved. The result of model prediction and experimental verification were discussed. The result indicated that the model established in this work was representative and could guide the in-situ consolidation process to control the void content. Since the purpose of this work was to explore the relationship between process parameters and the void content, thus the influence of void content on mechanical properties was not further discussed.

7 During the in-situ consolidation process, the prepreg and the substrate are melted by the heat source first, and then consolidated by the compaction roller. Why?

Reply: Thank you for your comments. The description here in the original manuscript was inaccurate. The accurate description should be “During the in-situ consolidation process, the prepreg and the substrate are melted by the heat source first, bonded together by the compaction roller, and then the bonded layers are cooled and crystallized.” We have modified the description.

8 The literature should be strengthen using recent publication in Polymers.

Reply: Thank you for your comments. The literature review has been strengthen by the lasted publication in Polymers. These references included Ref [1-5], [16-18] and [29].

Automated fiber placement (AFP) In-situ consolidation of continuous CF/high performance thermoplastic composite (Figure 1)[1-5], which is based on continuous fiber-reinforced thermoplastic prepreg, positioning, laying-up and consolidation by automatic placement equipment according to the requirements of the mathematical model, components are manufactured in the thickness direction by layer accumulation, and the manufacturing is accomplished when the thickness reaches the designed value, thus the traditional manufacturing process of first laying-up and then curing in the autoclave is no longer needed[6-12].

Zhu Liu[29] established a microscale unit cell with random distribution of fibers, interfaces and voids based on the random sequential adsorption algorithm to investigate The quantitative effects of voids content on strength and modulus under the loading of transverse tension.

9 Support the introduction using; fabrication techniques of polymeric nanocomposites: a comprehensive review, modeling of drilling process of gfrp composite using a hybrid random vector functional link network/parasitism-predation algorithm, effect of surface preparation on the strength of vibration welded butt joint made from pbt composite.

Reply: Thank you for your comments. These three articles have been used to support the Section 1 Introduction as recommended. As seen in Ref [13-15].

The thermoplastic composite has outstanding characteristics which are different from the traditional thermosetting composite, such as high toughness, high strength, fast forming, and recyclability[13-15].

10 A void dynamic model and boundary conditions during the process of in-situ consolidation were established based on the two-dimensional Newtonian fluid squeeze flow model. The cyclic iteration method was used to calculate the variety of the void content during the process of in-situ consolidation by combining the heat transfer model with the void dynamic model. This claim should be discussed.

Reply: Thank you for your comments. The description here in the original manuscript was inaccurate. In this work, the void dynamic model and boundary conditions during the process of in-situ consolidation was not combined with the heat transfer model. The result of the heat transfer model was only used as the input parameter of the void dynamics model. Thus, the heat transfer model was not described. We have modified the description in the manuscript.

A void dynamic model and boundary conditions during the process of in-situ consolidation were established based on the two-dimensional Newtonian fluid squeeze flow model. The cyclic iteration method was used to predict the variety of the void content during the process of in-situ consolidation.

Reviewer 3 Report

Dear Author(s),

The manuscript Polymers 2022, 14 and titled “Research on Void Dynamics during In-situ Consolidation of 2 CF/High Performance Thermoplastic Composites” was reviewed. It is good paper.

The research aim is to develop a model to define the dynamic void content during processing. Paper is well written and it was clearly explained for reader stand points. Findings from the research were acceptably discussed considering the literature, and conclusion was consistent with generated data. 

It was evaluated that it is interesting paper and finding could attact readers. The manuscript can be accepted after some revisions.

  1. Please explain the strength mechanism in interlaminar region especially considering the void geometry.

  1. Please explain the bonding mechanism in interlaminar region providing a sketche drawing.

My best regards,

Author Response

Dear editor and reviewers:

Thank you for your comments. These comments help us to improve the quality of manuscripts. We have modified the manuscript according to each suggestion, and highlighted the modified part in the manuscript. Let's list the responses to each comment.

Thank you for your help on this manuscript.

Best wishes

Qinghua Song

Reply to Reviewers:

The manuscript Polymers 2022, 14 and titled “Research on Void Dynamics during In-situ Consolidation of 2 CF/High Performance Thermoplastic Composites” was reviewed. It is good paper.

The research aim is to develop a model to define the dynamic void content during processing. Paper is well written and it was clearly explained for reader stand points. Findings from the research were acceptably discussed considering the literature, and conclusion was consistent with generated data.

It was evaluated that it is interesting paper and finding could attact readers. The manuscript can be accepted after some revisions.

1 Please explain the strength mechanism in interlaminar region especially considering the void geometry.

Reply: Thank you for your comments. During the process of in-situ consolidation, the interlaminar strength of thermoplastic composites is mainly determined by the fusing bonding between layers. Simultaneously, the degree of fusing bonding is related to the heating temperature, compaction pressure, dwell time and void dynamic discussed in this paper.

For semi-crystalline thermoplastic composites, when the temperature is above the glass transition temperature (for the crystal region, the temperature needs to be higher than the melting temperature), the molecular chains can diffuse between layers. The traditional creep theory holds that the diffusion distance of molecular chains increases with time, expressed as follows:

                                ï¼ˆ1)

Where  is the diffusion distance of molecular chains,  is diffusion time, is molecular weight.

According to Kim[1], the bonding strength is in proportion to the diffusion distance of molecular chains:

                               ï¼ˆ2)

Where  is the bonding strength.

Thus, with the increase of heating temperature, compaction pressure and dwell time, the bonding strength increases. However, the voids would hinder the diffusion of the molecular chains and reduces the bonding strength. Hence, the void content should be reduced as much as possible during the in-situ consolidation process.

Ref [1] KIM Y H, WOOL R P. Theory of healing at a polymer-polymer interface[J]. Macromolecules, 1983, 16(7): 1115-1120.

2 Please explain the bonding mechanism in interlaminar region providing a sketche drawing.

Reply: Thank you for your comments. Due to the existence of surface roughness of continuous fiber reinforced thermoplastic prepreg, the micro morphology of the prepreg surface geometry is irregular. Therefore, during the initial stage of in-situ consolidation, the surfaces of adjacent two-layers are not in full contact, but the degree of contact between the two-layers changes gradually with the heating of laser source and the compaction roller. After the two adjacent layers are in intimate contact, the molecular chains begin to fuse. The bonding strength between layers is determined by the fusing bonding. The development of fusing bonding between layers during in-situ consolidation process could be summarized, as follow in Figure 1.

Fig.1 The development of fusing bonding between layers during in-situ consolidation process

Although the bonding strength is affected by the interlaminar void, however, this paper focuses on the void dynamics during the in-situ consolidation process. The effect of void on the interlaminar strength is not the main goal of this paper. Therefore, there is no more description about the development of interlaminar strength in the manuscript.
